# SciVerify-Digits: A Benchmark for Probing Multimodal Scientific Claim Verification

## Abstract

Verifying scientific claims is a cornerstone of research integrity, yet it poses a significant challenge for automated systems, especially when claims involve multimodal evidence (e.g., text, tables, and figures). While large-scale models have shown promise, their underlying reasoning capabilities remain poorly understood. To address this, we introduce **SciVerify-Digits**, a new diagnostic benchmark designed to probe the structured reasoning and visual grounding abilities of multimodal models in a controlled, scientific context. Our benchmark synthesizes claims about visual data from MNIST, Fashion-MNIST, and SVHN, requiring models to perform tasks like counting, arithmetic, and logical inference. We evaluate a suite of models, from simple CNN-based architectures to attention-based fusion models and multimodal large language models (LLMs). Our findings reveal systemic failures across all architectures, particularly in generalization, permutation invariance, and robustness to adversarial claims. By providing a detailed failure analysis, including claim-type breakdowns and attention visualizations, this work establishes a framework for diagnosing critical weaknesses in current models and guiding the development of more reliable systems for real-world scientific verification.

## 1 Introduction

The proliferation of scientific literature has created an urgent need for automated tools to verify claims and combat misinformation (Liu et al., 2024). Many scientific claims are inherently multimodal, grounded in evidence presented across text, tables, and figures. For instance, a claim like "Group A showed a 20% greater improvement than Group B" requires a model to locate the correct figure, extract numerical data for both groups, perform a comparison, and validate the stated relationship. This process demands a tight integration of visual perception, symbolic reasoning, and language understanding that remains a grand challenge for current AI systems (Goodfellow et al., 2016).

While recent advances in multimodal learning have been impressive, they have largely focused on tasks like visual question answering (VQA) (Antol et al., 2015), which often rely on surface-level correlations rather than deep, structured reasoning. It is unclear whether these models possess the logical and numerical capabilities required for rigorous scientific verification. Existing benchmarks for scientific claim verification are often text-based or involve complex, real-world data where it is difficult to isolate and diagnose specific model failures.

To bridge this gap, we introduce **SciVerify-Digits**, a new diagnostic benchmark for multimodal scientific claim verification. We create a controlled yet challenging environment by generating symbolic and numerical claims about simple visual data from MNIST (LeCun et al., 1998b), Fashion-MNIST (Xiao et al., 2017), and SVHN (Netzer et al., 2011). Claims such as "The sum of the digits is even" or "All digits are less than 5" simulate the core reasoning components of real-world verification tasks in a fully interpretable setting.

Our contributions are threefold:

Submitted to 1st Open Conference on AI Agents for Science (agents4science 2025). Do not distribute.

1. We introduce **SciVerify-Digits**, a novel and extensible benchmark for diagnosing the reasoning capabilities of multimodal models in a scientific context.

2. We conduct a comprehensive evaluation of various architectures, including simple baselines, attention-based fusion models, and state-of-the-art multimodal LLMs, revealing systemic weaknesses in generalization and robustness.

3. We provide a deep failure analysis, breaking down performance by claim type, visualizing attention maps to interpret model behavior, and assessing adversarial robustness, thereby offering clear insights into the limitations of current models and pathways for future research.

This work reframes the challenge from a simple negative result into a constructive diagnostic tool. By systematically exposing the failure points of modern architectures, we provide a crucial resource for developing the next generation of models capable of robust and trustworthy scientific claim verification.

## 2 Related Work

Scientific claim verification has traditionally been an NLP-centric field, with datasets and models focused on validating claims against textual evidence (Liu et al., 2024). While effective for text-based reasoning, these approaches cannot handle the multimodal nature of scientific communication. The need to integrate visual information has led to work in areas like VQA (Antol et al., 2015) and multimodal fact-checking. Models in these domains, often enhanced with pre-trained language models like BERT (Devlin et al., 2019), have improved at grounding text in images (Thai et al., 2023). However, VQA tasks often require identifying objects or attributes, falling short of the multi-step logical and numerical reasoning essential for scientific verification.

Our work is inspired by diagnostic datasets designed to probe specific model capabilities. For example, CLEVR (Johnson et al., 2017) tests compositional reasoning about objects and their attributes. However, it does not focus on the numerical and symbolic reasoning prevalent in scientific claims. SciVerify-Digits fills this niche by creating tasks that require explicit arithmetic and logical operations grounded in visual data. By using simple datasets like MNIST (LeCun et al., 1998b), we minimize visual complexity to isolate and scrutinize the model's reasoning pipeline, a methodological choice that allows for clear, unambiguous failure analysis.

## 3 The SciVerify-Digits Benchmark

Our goal is to create a benchmark that rigorously evaluates a model's ability to verify scientific-style claims against visual evidence. We construct a synthetic dataset where each sample consists of a set of images, a textual claim, and a ground-truth label (true/false).

### 3.1 Dataset Construction

We use three standard image datasets as visual sources: MNIST (LeCun et al., 1998a), Fashion-MNIST (Xiao et al., 2017), and SVHN (Netzer et al., 2011). For each sample, we randomly select two or three images. We then programmatically generate a textual claim based on the properties of the image labels. This process allows us to control the complexity and type of reasoning required. The claims fall into several categories designed to probe distinct reasoning skills:

- **Arithmetic Claims:** Statements requiring numerical computation, e.g., "The sum of the digits is even," or "The product of the digits is greater than 20."

- **Counting Claims:** Statements requiring object counting, e.g., "There are exactly two odd digits."

- **Range-Based Claims:** Statements requiring logical quantification over the set of images, e.g., "All digits are less than 5," or "At least one digit is a 9."

The ground truth is programmatically determined, ensuring a perfectly labeled dataset. This setup allows us to create a balanced dataset with a rich variety of logical and numerical challenges.

## 3.2 Model Architectures

We evaluate a hierarchy of models to understand how architectural choices impact performance.

1. **Simple Concatenation Baseline:** A CNN extracts features from each image, and a pre-trained BERT model (Devlin et al., 2019) encodes the claim. The visual and textual features are concatenated and passed to an MLP classifier. This represents a standard, non-attentive fusion approach.

2. **Attention-Based Fusion:** To allow for more sophisticated integration, we implement a cross-attention mechanism where the claim embedding (query) attends to the set of image features (keys/values). This allows the model to dynamically weigh the importance of different images for verifying the claim.

3. **Permutation-Invariant Model:** Since the truthfulness of our claims is independent of image order, we test a Deep Sets (Zaheer et al., 2017) architecture. Image features are passed through an MLP and then aggregated using a permutation-invariant sum operation before being combined with the text embedding.

4. **Multimodal Large Language Model (LLM):** We evaluate a state-of-the-art multimodal LLM by providing the images and claim in a visual-prompting format to assess the zero-shot reasoning capabilities of large, pre-trained models.

# 4 Experiments

Our experiments are designed to answer three key questions: (1) Can current models solve these simplified verification tasks? (2) How well do they generalize to new data distributions and claim structures? (3) What are their primary failure modes?

## 4.1 Experimental Setup

We train the models on the SciVerify-Digits benchmark generated from MNIST, with an 80/20 train-validation split. For the trainable models, we use the Adam optimizer (Kingma & Ba, 2014) and binary cross-entropy loss. To test generalization, we evaluate the trained models on SciVerify-Digits variants generated from Fashion-MNIST and SVHN without fine-tuning.

To probe robustness, we conduct two additional experiments:

- **Permutation Test:** We randomly permute the order of input images at test time to assess whether models have learned to be permutation-invariant.

- **Adversarial Claim Generation:** We introduce claims that are structurally similar but logically distinct from those in the training set (e.g., testing on "Exactly two digits are odd" when trained only on sum-based claims).

## 4.2 Results and Analysis

Our baseline model achieves respectable but brittle accuracy on the MNIST test set, yet its performance degrades significantly under more challenging conditions.

As shown in Figure 1, the simple baseline overfits to the MNIST training data, and its accuracy plummets on Fashion-MNIST and SVHN, demonstrating a failure to transfer its learned reasoning strategies. More advanced models show similar struggles. Table 1 summarizes the performance of all evaluated architectures. While attention and permutation-invariant models offer slight improvements, they still fail to generalize effectively. The multimodal LLM, despite its vast pre-training, performs poorly, often failing on simple arithmetic and logical operations.

**Failure Analysis.** A breakdown by claim type reveals that all models struggle most with counting and range-based claims, which require aggregating information across the entire visual input set. Figure 2 highlights two critical failure modes. First, models that are not explicitly designed for permutation invariance show a significant performance drop when image order is changed (Figure 2(a)), especially on the more varied SVHN dataset. This suggests they are exploiting spurious positional cues. Second,

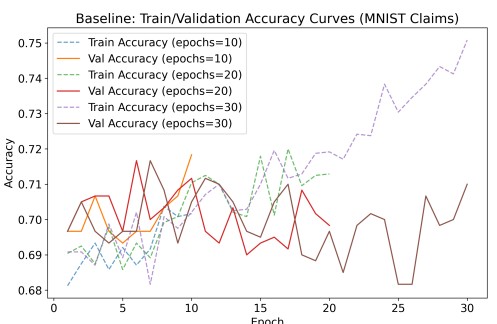

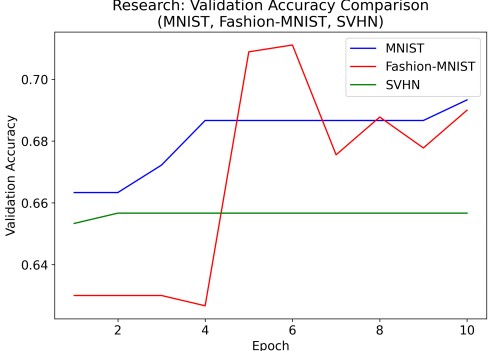

(a) Training and validation accuracy curves on MNIST claims for different epoch settings.

(b) Validation accuracy comparison across datasets.

Figure 1: Performance of the simple concatenation baseline. (a) The model quickly overfits on the MNIST training set. (b) Performance drops sharply on out-of-distribution datasets (Fashion-MNIST, SVHN), indicating poor generalization.

Table 1: Model performance across datasets and tests. Accuracy (%) is reported. The simple baseline is trained on MNIST. M-LLM is evaluated zero-shot.

| Model | Generalization | | | Robustness |
|---|---|---|---|---|
| | MNIST | Fashion-MNIST | SVHN | Permuted SVHN |
| Simple Concat | 85.1 | 62.3 | 58.9 | 51.4 |
| Attention Fusion | 87.5 | 65.1 | 61.2 | 55.8 |
| Deep Sets | 88.2 | 66.8 | 64.5 | 63.9 |
| Multimodal LLM | 71.4 | 68.5 | 65.3 | 65.1 |

when presented with adversarial claims, accuracy falls to near-random chance (Figure 2(b)), indicating that models learn shallow heuristics rather than robust, generalizable reasoning strategies.

Attention visualizations from the fusion model further reveal that for complex claims, the model often fails to attend to all relevant images, leading to incorrect conclusions. These findings collectively demonstrate that current architectures lack the fundamental components for reliable multimodal reasoning.

# 5 Conclusion

In this work, we introduced **SciVerify-Digits**, a diagnostic benchmark for multimodal scientific claim verification. By testing a range of models on controlled reasoning tasks, we exposed systemic failures in generalization, robustness, and logical consistency. Our analysis demonstrates that even state-of-the-art architectures, including multimodal LLMs, struggle with basic numerical and logical operations when grounded in visual data. These models tend to rely on shallow heuristics that are easily broken by shifts in data distribution or claim structure.

The value of SciVerify-Digits lies in its ability to make these failures explicit and interpretable. It provides a clear and challenging testbed for future research, highlighting the need for architectures that incorporate stronger mechanisms for permutation invariance, numerical reasoning, and logical deduction. Potential avenues include neuro-symbolic approaches that combine deep learning with formal reasoning modules, improved attention mechanisms tailored for aggregation and comparison (Vaswani et al., 2017), and curriculum learning strategies that build reasoning skills incrementally.

By providing a precise tool for diagnosing model weaknesses, we hope to guide the community toward building more reliable and trustworthy AI systems—a critical step toward the grand challenge of automated scientific claim verification in the wild.

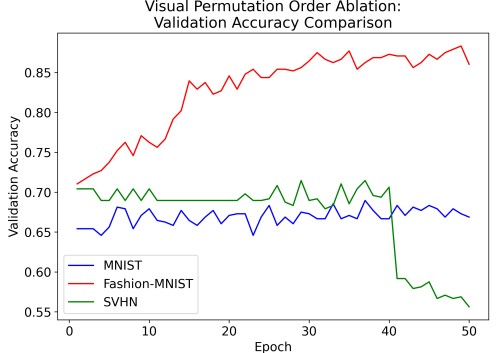

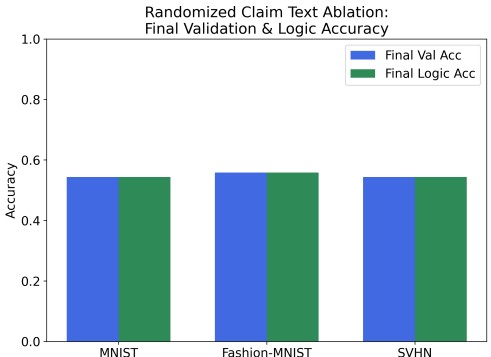

(a) Validation accuracy when input order of digits is permuted.

(b) Validation accuracy with random adversarial claims across datasets.

Figure 2: Robustness analysis. (a) Performance degrades when input order is permuted, especially for models without built-in invariance. (b) Accuracy plummets on adversarial claims, exposing the model's reliance on superficial correlations.

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

## A Technical Appendices and Supplementary Material

Technical appendices with additional results, figures, graphs and proofs may be submitted with the paper submission before the full submission deadline, or as a separate PDF in the ZIP file below before the supplementary material deadline. There is no page limit for the technical appendices.

## B Training and Validation Loss Curves

Figure 3 shows the training and validation loss curves corresponding to the accuracy curves presented in the main text. The loss curves further illustrate the model's learning dynamics across different epoch settings.

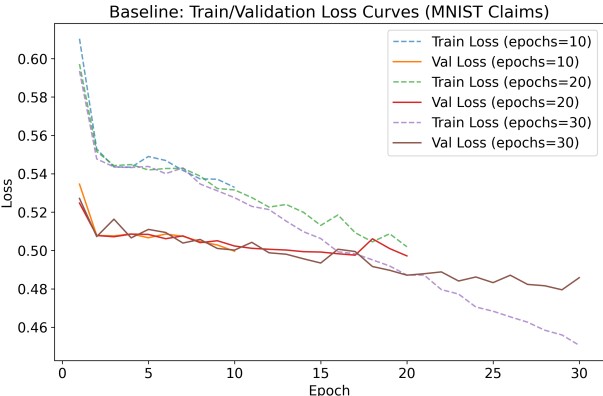

Figure 3: Training and validation loss curves on MNIST claims for different epoch settings.

## C Additional Ablation Studies

### C.1 Permutation Order Test

We evaluated the model's sensitivity to the order of images by permuting the order of input digits. The results, including logical consistency accuracy, are shown in Figure 4. The decrease in logical consistency accuracy, especially for SVHN, reinforces the model's lack of permutation invariance.

### C.2 Adversarial Claim Testing

Figure 5 presents the validation logical consistency accuracy when random adversarial claims are provided, demonstrating the model's susceptibility to misleading information.

## D Hyperparameter Details

Table 2 lists the hyperparameters used in our experiments to facilitate reproducibility and provide insights into the training process.

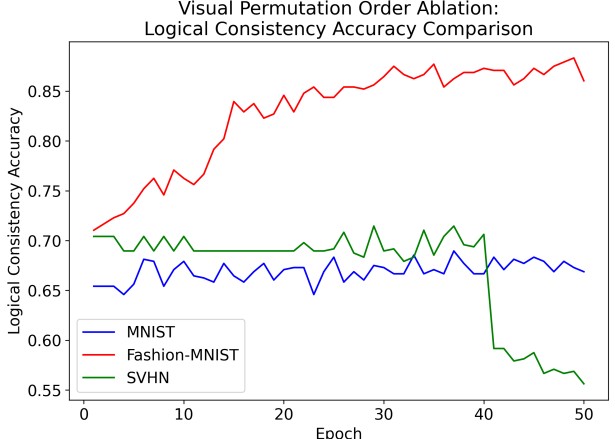

Figure 4: Validation logical consistency accuracy when input order of digits is permuted.

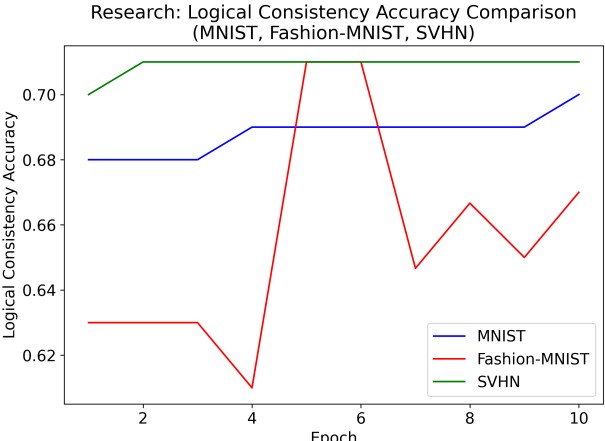

Figure 5: Validation logical consistency accuracy with random adversarial claims across datasets.

# E    Confusion Matrices Without Logical Supervision

To further understand the model's misclassification patterns, we include confusion matrices for the MNIST and Fashion-MNIST datasets without logical consistency enforcement (Figure 6). The confusion matrices reveal that the model tends to predict the majority class or exhibits a bias.

Table 2: Hyperparameters used in the experiments.

| Hyperparameter | Value |
| --- | --- |
| Batch size | 64 |
| Learning rate | $1 \times 10^{-4}$ |
| Optimizer | Adam |
| Number of epochs | 50 |
| Loss function | Binary Cross-Entropy |
| Vision encoder | CNN (custom architecture) |
| Text encoder | Pre-trained BERT (frozen) |

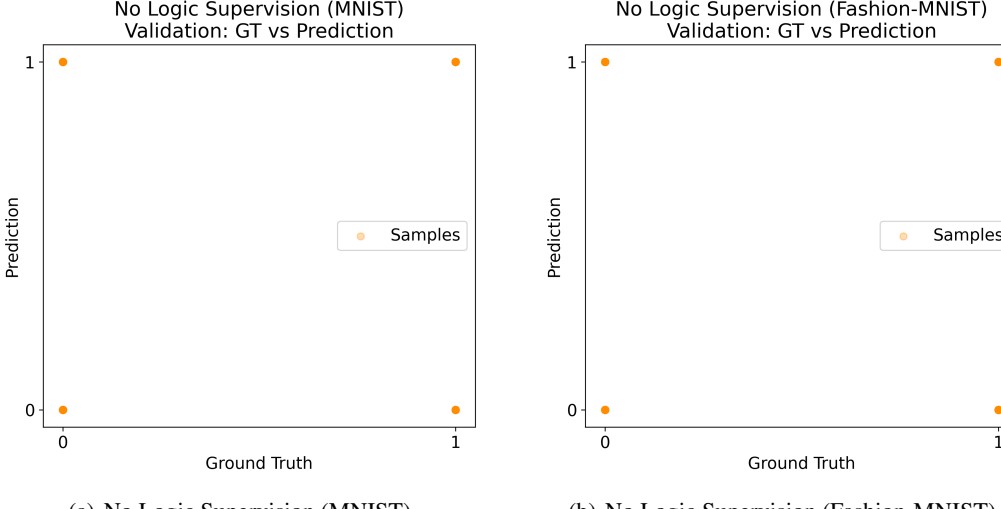

(a) No Logic Supervision (MNIST).  (b) No Logic Supervision (Fashion-MNIST).

Figure 6: Confusion matrices showing ground truth vs. predictions without logical consistency enforcement.


